# Three-Dimensional Human Posture Recognition by Extremity Angle Estimation with Minimal IMU Sensor

**DOI:** 10.3390/s24134306

**Published:** 2024-07-02

**Authors:** Yaojung Shiao, Guan-Yu Chen, Thang Hoang

**Affiliations:** 1Department of Vehicle Engineering, National Taipei University of Technology, Taipei 106344, Taiwan; 2Railway Vehicle Research Center, National Taipei University of Technology, Taipei 106344, Taiwan; 3Faculty of Transportation Mechanical Engineering, The University of Danang-University of Science and Technology, Danang 550000, Vietnam

**Keywords:** attitude recognition, inertial measurement unit (IMU), spatial coordinates, joint angle estimation

## Abstract

Recently, posture recognition technology has advanced rapidly. Herein, we present a novel posture angle calculation system utilizing a single inertial measurement unit and a spatial geometric equation to accurately identify the three-dimensional (3D) motion angles and postures of both the upper and lower limbs of the human body. This wearable system facilitates continuous monitoring of body movements without the spatial limitations or occlusion issues associated with camera-based methods. This posture-recognition system has many benefits. Providing precise posture change information helps users assess the accuracy of their movements, prevent sports injuries, and enhance sports performance. This system employs a single inertial sensor, coupled with a filtering mechanism, to calculate the sensor’s trajectory and coordinates in 3D space. Subsequently, the spatial geometry equation devised herein accurately computed the joint angles for changing body postures. To validate its effectiveness, the joint angles estimated from the proposed system were compared with those from dual inertial sensors and image recognition technology. The joint angle discrepancies for this system were within 10° and 5° when compared with dual inertial sensors and image recognition technology, respectively. Such reliability and accuracy of the proposed angle estimation system make it a valuable reference for assessing joint angles.

## 1. Introduction

Human posture-recognition systems have attracted significant attention and advancements in recent years. This technology is important in healthcare, sports, gaming, and security [1,2,3,4]. These systems use devices that can track movement patterns, measure changes in posture and joint angles, and analyze data in real time to meet exercise and recovery needs. Human posture recognition has a wide range of uses, including capturing human body movements in movies, monitoring home care, training athletes, and assisting in post-accident recuperation [5,6,7,8]. In order to accurately monitor and examine human posture, it is crucial to possess a comprehensive knowledge of human anatomical terminology and measurements. The nomenclature of the human body consists of standardized names used to describe its many parts and structures. The worldwide accepted system for this purpose is called Terminologia Anatomica (TA). Furthermore, anthropometry encompasses the quantification of physical attributes such as stature, mass, and bodily girths (e.g., waist, chest), which are essential for evaluating body ratios and well-being. Precise data gathering is enhanced by the use of instruments such as scales, rulers, and imaging techniques such as MRI and CT scans [9,10,11].

Motion capture technology, often known as motion capture, is a technique used to monitor and record human movement activities. Originally used by McDonald to improve animation effects by recording performers’ motions, modern motion capture technologies mostly depend on optical tracking techniques. Vivid markers (or dots) on the actor’s body bounce light back to the cameras and capture their locations [12]. Picture data from several cameras are then analyzed to determine the differences in the markers on the X, Y, and Z axes. Kinect is a popular choice among the several types of cameras used to detect human motion. Image recognition is one of the most common applications in this field. This study discusses the extensive usage of Kinect in several areas, including research, medical applications, entertainment, and human–computer interaction [13,14,15,16]. It evaluated several methods for recognizing gestures in Kinect to determine the most effective method in terms of precision and speed [17].

With advances in electrical technology, inertial measurement units (IMU) have become smaller and more stable. This development enhances the overall efficiency and adaptability of human posture-recognition technology [18,19,20,21,22,23]. In addition to utilizing Kinect cameras to analyze human posture, the use of IMUs has seen significant advancements in research conducted at universities [24,25,26]. Bellitti et al. [27] used an inertial measurement unit combined with a stretch sensor to design a low-cost knee-joint ligament relaxation system. In this system, two inertial sensors were placed on the upper and lower ends of the knee and connected using a stretch sensor. A gyroscope, accelerometer, and magnetometer were combined to obtain the attitude information of the knee ligament. Kim et al. [28] used the OpiTrack sportswear device of the Xsens IMU to record the posture information of the human body with six inertial sensors at 30 Hz and estimated the joint position and rotation angle through the bidirectional recurrent neural network of ConvLSTM. A large IMU database is required for training. The experimental results showed that each joint position had an error of approximately 50 mm, and the joint angle had an average error of 11.31°. McGrath et al. [29] used a wearable IMU to estimate the angles of human hip and knee joints while walking on a treadmill. The experimental results showed that the internal and external rotation angles of the hip joint were quite different, and it was found that the movement of the human soft tissue affects the accuracy of the sensor. Slyper et al. [30] placed five inertial measurement units on clothes and used the acceleration value read to match the acceleration value in the dynamic capture database to find the most suitable action and project it on the screen. However, they cannot provide a real-time representation of the tester’s posture. Huang et al. [31] designed a rehabilitation effectiveness monitoring system using a si*x*-axis inertial sensor combined with a sensor-fusion algorithm, which was primarily used to monitor the rehabilitation actions of patients with Parkinson’s disease. Ten inertial sensors were placed on the patient’s body, specifically on the upper extremities and forearms, femur and crus, thorax, and abdomen. The patient then participated in the Lee Silverman Voice Treatment Big (LSVT BIG) rehabilitation course. Data on the body postures of various parts during the rehabilitation course were collected and applied to 3D character objects on the Unity 3D game platform. The different postures and movements of the patient’s body were then simulated, and the characteristic angles of the limbs were calculated through the posture inclination angles. These characteristic angles allowed for viewing the patient’s movements during the rehabilitation course and continuous tracking of the rehabilitation progress, effectively providing physicians with the ability to judge the patient’s rehabilitation effect. However, the disadvantage is that ten inertial sensors must be placed on the patient, which causes discomfort and incurs high costs.

Utilizing the Kinect camera and IMU together aided in validating the kinematics. Banos et al. [32] investigated the transferability of activity detection in an Internet of Things (IoT) sensor environment using IMU and Kinect sensors. This study demonstrated the effective translation of data from IMUs recording three-dimensional acceleration on various body segments to Kinect sensors recording three-dimensional locations, resulting in high levels of precision. Tian et al. [33] proposed the use of an inertial sensor with an unscented Kalman filter (UKF) for sensor fusion to increase signal stability and avoid signal loss. In the experiment, the IMU and Kinect were used to capture the attitude information. Combining an inertial sensor with a physical geometry constraint method can yield a better tracking effect and provide stable position information. One disadvantage is the error caused by the use of additional inertial sensors.

In the literature discussed above, to estimate the joint angles of the body and limbs, more than two IMUs were required to collect attitude signals, increasing the cost of use and reducing user comfort. Therefore, the goal of this study is to use a minimum number of IMUs to estimate the joint angles of the body and limbs. Shiao et al. [34] used an IMU to track the position of the object. Assuming that the arm is projected on the XY plane, a set of mathematics is established using the relevant geometry model; pixels are used as units to simulate joint positions, and then camera image recognition technology and a spatial pyramid matching algorithm are used for verification. However, that research is only for 2D motion recognition, so this research will establish 3D motion identification technology. The objective of this study is to introduce an innovative system capable of identifying three-dimensional motion angles and postures of both the upper and lower human limbs using a single inertial measurement unit and a spatial geometric equation. The primary advantage of this wearable system is its ability to continuously monitor body movements without the typical constraints associated with camera-based methods, such as spatial limitations or occlusion.

## 2. The Material for Estimation by IMU

### 2.1. Extended Kalman Filter

A range of techniques were used to assess the precision of the angle measurement, including acceleration estimation, gyro estimation, Kalman filter (KF), complementary filter, gradient descent, and extended Kalman filter (EKF). Following our investigation, we determined that extended Kalman had the most favorable outcomes, as shown in Figure 1.

The EKF is more suitable for nonlinear problems than the KF, which is mainly used for general linear problems in discrete time-controlled linear systems. Many systems often exhibit nonlinear behavior. Consequently, this study uses an EKF. The EKF uses the KF on a model that is approximately linearized using the Jacobian series expansion of nonlinear functions. It then employs a KF to predict and update the measurement and state equations [29], as depicted in Figure 2.

The EKF algorithm comprises the following primary steps: Initially, the values of $x\left(0\right)$ and $p\left(0\right)$ are established, where $x\left(0\right)$ represents the initial estimation of the state and $p\left(0\right)$ represents the initial covariance of the error. Subsequently, the Jacobian matrix $w_{k-1}$ is computed based on the process model in relation to the state. During the time update step of the Kalman filter, the anticipated state estimate ${\hat{x}}_{k}^{-}$ is determined by combining the process model and the control input $u_{k-1}$. At the same time, the forecast error covariance $P_{k}^{-}$ is derived by using the Jacobian matrix F and the process noise covariance Q. The Jacobian matrix $H_{k}$ of the measurement model with respect to the state and the Jacobian matrix $v_{k}$ with respect to the noise are calculated thereafter. The Kalman gain $K_{k}$ is calculated by using the forecast error covariance $P_{k}^{-}$, the Jacobian matrix $H_{k}$, and the measurement noise covariance $R_{k}$. The revised state estimate ${\hat{x}}_{k/k}^{+}$ incorporates the measurement $z_{k}$. Ultimately, the state covariance ${\hat{P}}_{k/k}^{+}$ is revised. The procedure involves iterating through each time step and updating the state estimate and error covariance using the process model, measurement model, and their corresponding Jacobian matrices.

Figure 3 shows a schematic of the integration of sensors and estimation methods used in motion-tracking systems. The system is based on an EKF algorithm that processes data from three separate sensors: a gyroscope, an accelerometer, and a magnetometer. The sensors feed the data into the filter, which processes this information through several correction procedures to optimize the accuracy of motion tracking. The process began with an initial estimation of gravity, which was then refined using correction 1. Concurrently, the magnetic field was estimated and refined using correction 2. The refined estimates were combined to form an initial angular estimate. This initial estimate was further improved by correction 3, leading to the final output of the system: the angular position estimation. The result is represented as a 3-axis angle, which precisely indicates the orientation of the object being tracked in three-dimensional space. The figure illustrates a sophisticated algorithm that merges data from multiple sensors to track motion accurately, which is a critical capability for a wide range of applications in the technology and engineering domains.

### 2.2. Mathematical Model for Joint Angle Estimation



*(a) Estimation of upper extremity joint angles*



The joint angle estimation equation in this study was extended based on a mathematical model reported in the literature [34]. The angle of each body joint is estimated using spatial geometric equations. Figure 4 shows a schematic of the elbow joint angle estimation. First, the coordinates of the shoulder joint are set as (0, $l_{U1}+l_{U2}$). Equations (1) and (2) can be readily expressed with centers located at (0, $l_{U1}+l_{U2}$) and ($A_{U},B_{U})$ correspondingly. The solution for the intersection of these two circles represents the precise location of the elbow joint. Furthermore, the validity of this solution is supported by the equation for a straight line, (3). The inverse trigonometric function, the radian R calculated from Formula (4) can be brought into Formula (5) to estimate the angle between the upper arm and the body (shoulder joint angle), and finally, $\theta_{U1}$ the elbow joint angle can be deduced according to Formula (6). The meanings of the symbols are explained in Table 1.
(1)$${x_{U}}^{2}+{\left(y_{U}-\left(l_{U1}+l_{U2}\right)\right)}^{2}={l_{U1}}^{2}$$
(2)$${\left(x_{U}-a_{U}\right)}^{2}+{\left(y_{U}-b_{U}\right)}^{2}={l_{U2}}^{2}$$
(3)$$y_{U}=ax_{U}$$
(4)$$arctan\frac{\left|(x_{U}-0)\right|}{\left|(y_{U}-\left(l_{U1}+l_{U2}\right))\right|}$$
(5)$$\theta_{U1}=R*(180/\pi )$$
(6)$$\theta_{U3}=180-(180-\theta_{U1}-\alpha )=\theta_{U1}+\alpha$$



*(b) Estimation of lower extremity joint angles*



Figure 5 shows a schematic of the knee joint angle estimation. First, the coordinates of the hip joint are set to (0, $l_{L1}+l_{L2}$), and the same solution is obtained according to the circle Equations (7) and (8), and the same solution is obtained from the linear Equation (9) to confirm the correct coordinate position of the knee joint. After obtaining the correct coordinates of the knee joint, use the inverse trigonometric function to calculate the radian R from Formula (10) and bring it into Formula (11) to calculate the angle between the thigh and the body (hip joint angle) $\theta_{L1}$, and finally deduce the knee joint angle according to Formula (12). The meanings of the symbols are explained in Table 2.
(7)$${x_{L}}^{2}+{\left(y_{L}-(l_{L1}+l_{L2})\right)}^{2}={l_{L1}}^{2}$$
(8)$${\left(x_{L}-a_{L}\right)}^{2}+{\left(y_{L}-b_{L}\right)}^{2}={l_{L2}}^{2}$$
(9)$$y_{L}=ax_{L}+b$$
(10)$$arctan\frac{\left|(x_{L}-0)\right|}{{|(y}_{L}-(l_{L1}+l_{L2}))|}=R$$
(11)$$\theta_{L1}=R*(180/\pi )$$
(12)$$\theta_{L3}=180-(180-\left(180-\theta_{L1}\right)-\beta ))=\;180+-\theta_{L1\_}\;\beta$$

### 2.3. Minimal IMU Model for Dynamic Motion Observation

To represent the movement posture of the human body fully, the human body undergoes a displacement phenomenon during movement. This research uses spatial geometry to obtain the angle; therefore, when the human body experiences a movement phenomenon, the displacement distance must be considered. Therefore, the displacement of the human body was obtained by using an inertial sensor on the chest, as shown in Figure 6. In the geometrical algorithm of hand space $x_{U}$, $y_{U}$ add the body displacement $z_{C}$, $y_{C}$, $y_{L}$ and add $z_{C}$, $y_{C}$ to the leg in the algorithm $x_{L}$, such as Formulas (13)–(18), so that the moving position can be estimated as a joint angle.
(13)$${\left(x_{U}+z_{C}\right)}^{2}+{\left(y_{U}+y_{C}-(l_{U1}+l_{U2})\right)}^{2}={l_{U1}}^{2}$$
(14)$${\left(x_{U}+z_{C}-a_{U}\right)}^{2}+{\left(y_{U}+y_{C}-b_{U}\right)}^{2}={l_{U2}}^{2}$$
(15)$$y_{U}+y_{C}=a(x_{U}+z_{C})+b$$
(16)$${\left(x_{L}+z_{C}\right)}^{2}+{\left(y_{L}+y_{C}-(l_{L1}+l_{L2})\right)}^{2}={l_{L1}}^{2}$$
(17)$${\left(x_{L}+z_{C}-a_{L}\right)}^{2}+{\left(y_{L}+y_{C}-b_{L}\right)}^{2}={l_{L2}}^{2}$$
(18)$$y_{L}+y_{C\;}=a(x_{L}+z_{C})+b$$

### 2.4. The Impact of IMU Placement Position

In previous studies, to estimate the joint angles of the upper limbs of the body, more than two IMUs were required to collect the posture signals of the upper limbs [35,36], which not only increased the cost of use but also caused poor user comfort. The unit is used to estimate the joint angle of the body limb; therefore, a relatively stable position must be selected to place the IMU. For the upper limb, when the upper arm of the human body moves, the lower arm must move accordingly. Therefore, we chose to place the inertial sensors on the wrist and forearm, as shown in Figure 7. In this study, acceleration was an important signal when estimating the joint angles. Therefore, five sets of flexing exercises were performed to measure acceleration values, as shown in Figure 7.

Because the hand of the upper extremity lifts upward during the flexing exercise, the rotation axis is the *X* axis with a smaller acceleration value; therefore, it is obtained from the *Y* and *Z* axes with larger acceleration values. After the experiment, it was found that placing the sensor on the wrist had a greater acceleration value than that on the forearm, as shown in Figure 8 and Figure 9, which is beneficial to the subsequent joint angle estimation; therefore, we chose to place the inertial sensor on the wrist. The vertical axis (A_Y) has units of “g,” where 1 g is equivalent to an acceleration of 9.8 m/s^2^, corresponding to the acceleration due to gravity. The horizontal axis represents time in seconds (s).

Prior research has shown that the collection of attitude signals to estimate the joint angles of the lower limbs of the body required the use of more than two IMUs [26]. For the lower body part, this study uses the leg stretching exercise to carry out the experiment. During the leg stretching exercise, the shank is mainly in motion; therefore, we chose to place inertial sensors on the ankle and shank and performed five sets of leg stretching movements as shown in Figure 10. Because the rotation axis of the leg-stretching movement is the *Z* axis with a smaller acceleration value, it is obtained from the *X* and *Y* axes with larger acceleration values, as shown in Figure 11 and Figure 12, respectively. After the experiment, it can be seen that when the sensor is placed on the ankle, the acceleration value of the *X* axis has a small signal fluctuation, and the acceleration value of the *Y* axis has a larger acceleration value, which is beneficial to the estimation of the joint angle later; therefore, the inertial sensor is selected, and the measuring device is placed at the ankle.

In this study, it was decided to place the upper part of the inertial measurement unit at the wrist and the lower part at the ankle. The hand placement direction is that the *X* axis is perpendicular to the arm and the *Y* axis is parallel to the arm; the leg placement direction is that the *X* axis is perpendicular to the shank and the *Y* axis is parallel to the shank. The body’s inertial sensor is placed on the chest because, when the human body rotates, there is mainly a large displacement.

The chest undergoes minimal movement, particularly the abdominal region. Therefore, positioning the inertial measurement unit (IMU) at this location yields more accurate signals with an optimal placement direction. It is recommended that the IMU be oriented along the *Z* axis perpendicular to the body and along the *Y* axis parallel to the body, as illustrated in Figure 13. Once the sensor position is determined, data such as the angular velocity, acceleration, and rotational angles along the three axes can be captured. Subsequently, an algorithm was employed to infer the motion trajectory. The primary objective of the experiment was to utilize a single inertial sensor to estimate the joint angles of specific body parts. Thus, a precise directional definition is crucial. Using the standing posture as a reference point, the alignment of the sensor direction with that of the wrist and ankle is essential. The *X*-axis acceleration defines the left-right swing direction (roll), the *Y*-axis acceleration denotes the rotation direction (yaw), and the *Z*-axis acceleration signifies the forward–backward swing direction (pitch). As gravity maintains a consistent direction, any change in attitude alters the gravity component. The attitude angle can be accurately calculated by integrating the transformation data with sensor fusion.

## 3. Motion Joint Angle Verification

### 3.1. Image Recognition Technology

A Kinect sensor is used in picture recognition technology to compare the joint angles determined by a solitary inertial sensor. Kinect is equipped with a 3D depth sensor and an infrared emitter that enables real-time tracking of the user in front of the camera. The random forest method, which is an effective ensemble learning technique, was used for classification and prediction purposes. Random forests can effectively categorize joints such as shoulders and elbows using bone data obtained from the Kinect camera in the context of recognizing significant locations on the human body. Random forests are renowned for their resilience and precision in managing intricate classification and prediction tasks. Ensemble learning is a strategy that integrates many decision trees to improve prediction accuracy. The random forests model constructs each decision tree individually, and the final forecast is determined by combining the predictions of all the individual trees. Random forests excel at efficiently processing large datasets and high-dimensional data. Random forests may enhance generalization performance and mitigate overfitting by using the technique of randomly selecting subsets of features and data points for each tree. This feature makes it especially well suited for applications such as picture identification, where the input data might be of great dimensionality and complexity. Random forests may use bone data obtained from the Kinect sensor to effectively categorize and forecast the positions of crucial joints, including the shoulders, elbows, and wrists. Significantly, the angle of the elbow joint undergoes a transition from approximately 90° to 30° during activity, as shown in Figure 14. The Random forests algorithm effectively captures and interprets tiny variations in joint angles during physical activities, as indicated by this range-of-motion detection.

### 3.2. The Difference between Image Recognition Technology and Single-Sensor Estimation of Joint Angle

The experimental methods included five groups each of flexing and five groups of leg stretching exercises. Figure 15 shows the angle of the elbow joint during curling. The angle measured during the preparatory movement was 90°. When the forearm is raised, the angle decreases as the shoulder approaches. The ange changed from 90 to 30 degrees, with an average difference rate of 5.8%. Figure 16 shows the knee joint angles measured during stretching exercises. The measured angle was 100° when the shank was raised and changed from 110° to 160° during further passive flexion. The average difference rate is 0.8%, and it can be seen that the joint angles obtained by the single inertial sensor and the Kinect sensor are similar.

The difference between the joint angles measured by a single inertial sensor and the Kinect image recognition technology was within 5°, and the angle changes were similar. Figure 17 and Figure 18 show the comparison, and the differences in the values can be clearly seen. The testing findings of elbow and knee motions demonstrate that the new posture angle calculation technique and the Kinect image recognition technology provide almost comparable measurements of joint angles. The elbow joint has a standard deviation of roughly 1.838 for the inaccuracy between the IMU and Kinect, with a variance of about 3.377. This suggests a strong correlation between the two measuring instruments, with negligible errors and few fluctuations. The IMU and Kinect measurements exhibit slight differences. The knee joint exhibits a standard deviation of roughly 1.062 and a variation of about 1.128 in the inaccuracy between the IMU and Kinect. Furthermore, this suggests a strong correlation between the two measuring instruments, with negligible errors and low fluctuations. The results obtained from the IMU and Kinect exhibit slight disparities.

Based on these findings, it is evident that the novel posture angle calculation technique used in this work may serve as a reliable benchmark for measuring joint angles. It exhibits a reasonably high level of accuracy and minimal mistakes when compared to the Kinect image recognition technology. The congruity between the data of the IMU and Kinect indicates that this novel technology has promise for extensive use in research and applications pertaining to the monitoring and analysis of bodily movements.

## 4. Experiment on Human Walking Posture

We use five IMU sensors to track human gait. Five inertial sensors were placed on both the wrists and ankles, and one IMU was placed on the chest to analyze the human walking motion for posture recognition in Figure 19. As there was no displacement of the chest during the walking experiment and the chest remained upright, no chest measurements were taken. The study focused on designing a human–machine interface to provide data on the angles of the shoulder, elbow, hip, and knee joints. This experiment focused on the posture and joint angles of the human body while walking in Figure 20. First, we confirmed the lengths of the upper arm $l_{U1}$, forearm $l_{U2}$, thigh $l_{L1}$, and shank $l_{L2}$ of the subject and then performed a walking exercise, as shown in Figure 21. The image depicts a sequence of figures illustrating the stages of bipedal walking. The series of six states arranged horizontally captures the progression from standing to walking and back to standing. Starting from the left, the first state portrays the “Standing” phase, showing a stick figure in an upright position with a circle representing the head, a vertical line for the body, and two straight lines for the legs. Moving to the right: The second state, labeled “Right swing phase-1”, demonstrates the initial movement of the right leg forward while the left leg remains stationary and vertical. The third state, labeled “Left swing phase-1”, mirrors the second state, with the left leg in motion and the right leg supporting the body. The fourth and fifth states repeat the right and left swing phases, respectively, indicating a continuous walking cycle. The sixth and final sate returns to the “Standing” phase, with the stick state standing upright and legs together. Throughout the states, the dashed blue lines trace the trajectory of the leg movement relative to the standing position. This schematic representation serves to analyze the mechanics of walking motion, which is commonly utilized in biomechanics to study the gait cycle.

By determining the lengths of the subject’s upper arms $l_{U1}$, arms $l_{U2}$, thighs $l_{L1}$, and calves $l_{L2}$ of the subject, along with the tester’s upper arm length of 26 cm, forearm length of 27 cm, thigh length of 36 cm, and shank length of 35 cm, we applied the coordinate algorithm to each inertial sensor. The trajectories of the wrist and ankle inertial sensors during walking, as depicted in Figure 22, provide the moving coordinates required to calculate the joint angles using spatial geometric equations. The colors of the points represent the positions of the points over time. As time progresses, the points change color, allowing us to visualize the movement or trajectory of the points in 3D space. This color-coding helps in understanding the sequence and progression of the points’ positions over the given period. Furthermore, its color changes continuously, so that each color is distinct from the previous one. This continuous color change makes it easy for the reader to follow and understand the path of the sensors during movement, allowing for a clear understanding of the movement path over time.

Figure 23 illustrates the variation in the joint angles throughout the walking motion. The shoulder joint angle remained close to 0° during the preparatory phase, and both directions were considered positive. During the exercise, the shoulder joint swung back and forth, reaching a maximum angle of 35°. Following the exercise, the shoulder joint angle returned to approximately 0°, transitioning from 0° to 35°.

At the initiation of the walking motion, the forearm is raised and moved in tandem with the upper arm in a swinging motion. Their hands remained in a consistent position while swinging back and forth. The angle of the elbow joint fluctuated during the swing and did not remain fixed at 90°. Consequently, the elbow joint angle varied from approximately 80° to 90°, returning to 180° at the conclusion of the movement.

Figure 24 depicts the evolution of joint angles during the walking motion. The hip joint angle remained close to 0° during the preparatory phase. Initially, the right foot was raised and reached a maximum angle of approximately 40°, followed by the left foot, which had a similar maximum angle. This alternating pattern continued, with the right foot lifting first, followed by the left foot. As the walking motion progressed, the hip joint angle returned to nearly 0° and transitioned during flexion from 0° to 40°. At the onset of the walking motion, the right foot was initially raised to a minimum joint angle of 153°, followed by the left foot reaching a minimum angle of 153°. Subsequently, the right foot was lifted first, followed by the left foot, and the knee joint angle reached 180° at the end of the walking motion. The knee angle transitioned from 180° to 153° during movement.

Based on the findings of the walking exercise experiment, it is evident that the shoulder joint angle oscillates back and forth, reaching a peak of approximately 35° during exercise. The elbow joint maintained a consistent posture, with an angle ranging between 80° and 90° throughout the exercise. When the thigh is lifted during movement, the hip joint angle is approximately 38°, and the knee joint angle reaches approximately 155° when the shank is raised. The data of Table 3 detailing the use of the IMU to measure the initial state prior to lifting the foot or hand and displaying the angular positions reveal several key observations and analyses. Initially, both the feet and hands exhibited angular values close to 0°, indicating their starting positions before any movement occurred. When the foot was raised, there were noticeable changes in both foot and body angles, highlighting significant variations in their angular values. Similarly, lifting the hand results in alterations in the angles of both the hand and body, showing the body’s response to the action of raising the hand. Motion angle graphs visually represent these angular changes during the process of lifting the foot or hand, offering a comprehensive view of the motion dynamics and reactions of the body. Overall, the data of Table 3 provided valuable insights into the angular adjustments of body parts during specific movements, thereby enhancing our understanding of body structure and motion patterns during various activities.

Our study has several limitations. Firstly, it requires five IMU sensors to track body movements, which can be cumbersome and inconvenient for practical applications. Secondly, while the system demonstrates relatively high accuracy, there are still inherent errors that may not suffice for applications requiring extremely high precision. Thirdly, the use of multiple sensors and complex algorithms increases the system’s cost and complexity. Lastly, the current system focuses on specific exercises and may not be scalable or easily integrated with other systems. Future research should aim to further reduce the number of sensors, enhance accuracy, optimize cost and complexity, and expand tracking and integration capabilities. Table 4 below provides a detailed analysis of the limitations of the IMU-based posture recognition system and discusses potential areas for improvement. Each limitation is described, followed by a discussion of its implications and suggestions for future research.

## 5. Conclusions

This study utilized the extended Kalman filter (EKF) to acquire precise attitude information, followed by the implementation of a novel attitude angle estimation system for estimating joint angles and attitude presentation. This system enables the estimation of body position using a single inertial sensor for one extremity, which is the objective of the study. The verification of the joint angles involves the use of dual inertial sensors and high-accuracy image recognition technology to cross-validate each other. The experimental results of the flexion and leg extension exercises demonstrated that the difference between the single inertial sensor and the dual inertial sensor was within 10°. Similarly, the variance between the single inertial sensor and the image recognition technology was within 5°, which closely approximated the actual motion angles. The user-friendly man-machine interface allows users to intuitively grasp their current posture and conveniently store all movement tracks while wearing the device. For athletes, immediate feedback on posture correctness and effectiveness can enhance training outcomes. Moreover, the system can be beneficial in medical settings for tracking user movements in a more private manner than using cameras, thereby avoiding blind spots during monitoring. In the rapidly evolving field of posture-recognition technology, a cutting-edge posture-angle calculation system leveraging a single inertial measurement unit (IMU) and spatial geometric equation was introduced. This wearable system facilitates continuous monitoring of body movements without the spatial limitations or occlusion issues commonly associated with camera-based methods. The advantages of this system are manifold, providing precise insights into posture changes to assist users in evaluating movement accuracy, preventing sports-related injuries, and optimizing sports performance. By integrating a single inertial sensor with a filtering mechanism for trajectory and coordinate calculations in 3D space in conjunction with the spatial geometry equation developed in this study, accurate joint angle computations can be achieved in response to changes in body posture. The effectiveness of the system was validated by comparing the estimated joint angles with those obtained using dual inertial sensors and image recognition technology. These outcomes underscore the reliability and precision of the new attitude angle estimation system, which serves as a valuable tool for joint angle assessment.

## Figures and Tables

**Figure 1 sensors-24-04306-f001:**
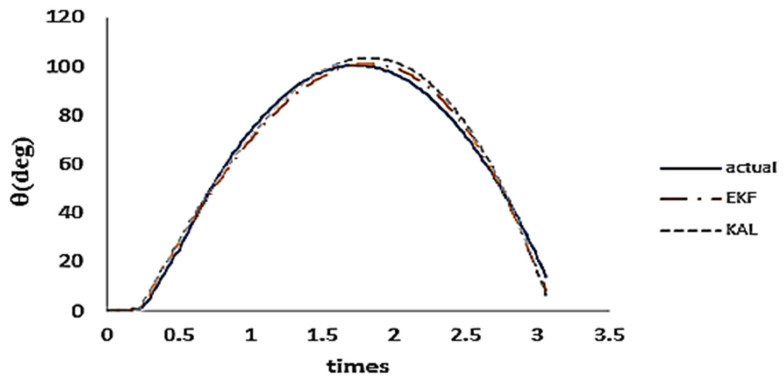
Comparison of curve errors among estimation algorithms.

**Figure 2 sensors-24-04306-f002:**
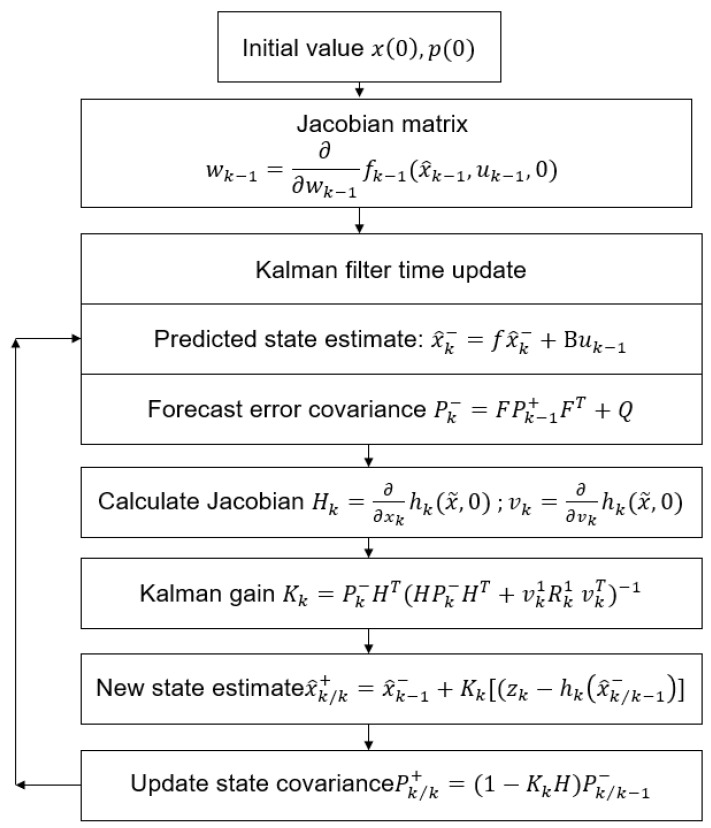
Diagram and formulas related to the use of Kalman filter in estimating the system state.

**Figure 3 sensors-24-04306-f003:**
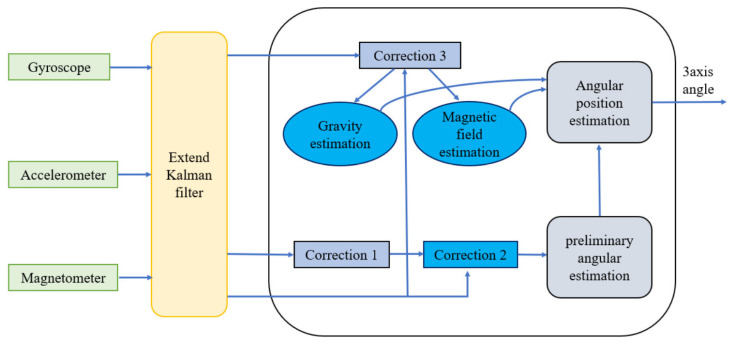
Sensor integration and estimation techniques in motion tracking systems.

**Figure 4 sensors-24-04306-f004:**
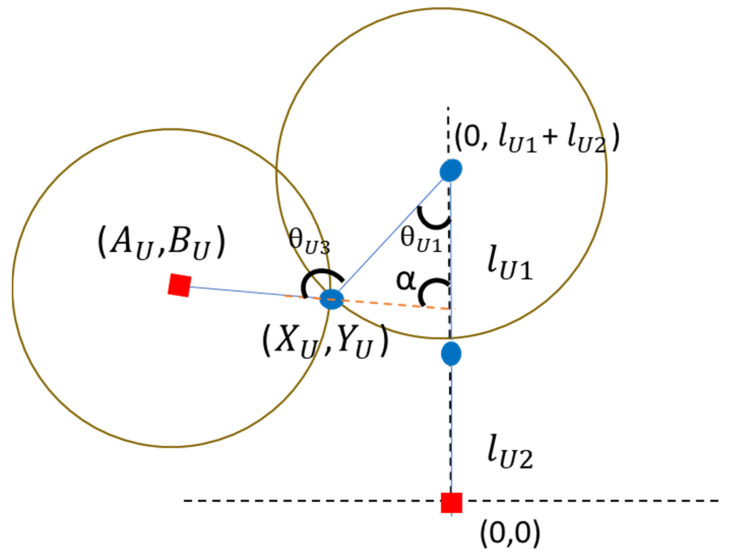
Diagram depicting the calculation of joint angles in the upper limb.

**Figure 5 sensors-24-04306-f005:**
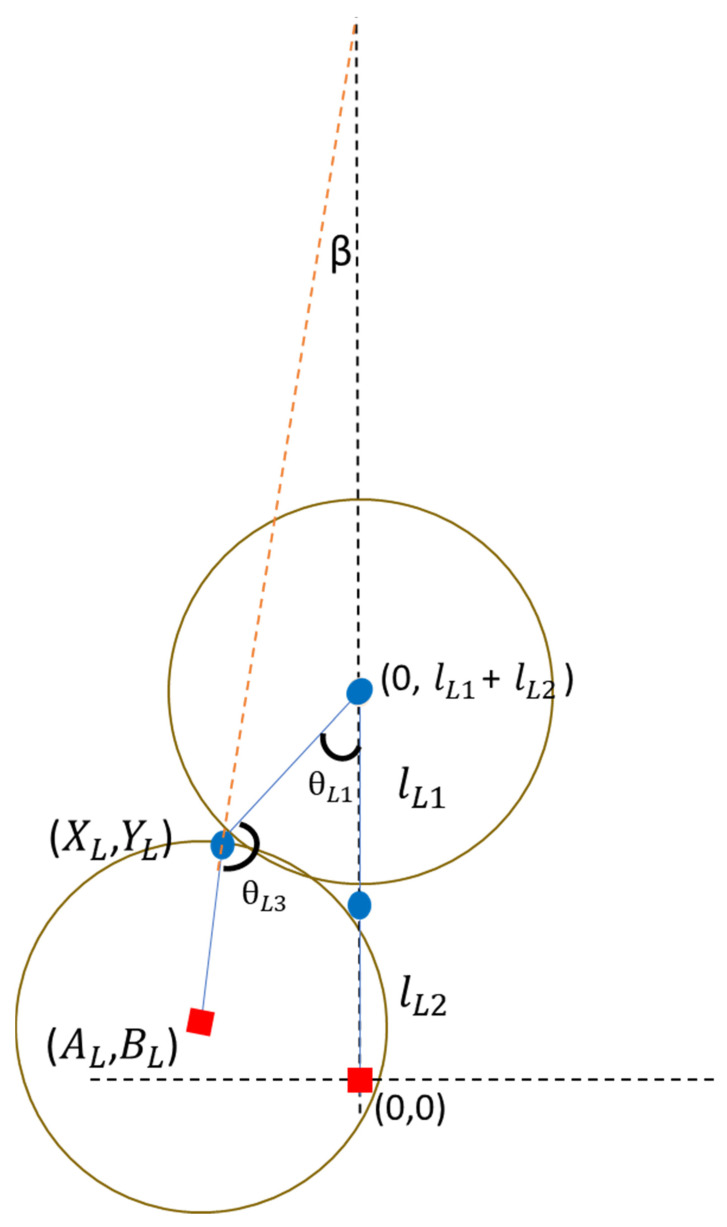
Schematic diagram for estimation of lower extremity joint angles.

**Figure 6 sensors-24-04306-f006:**
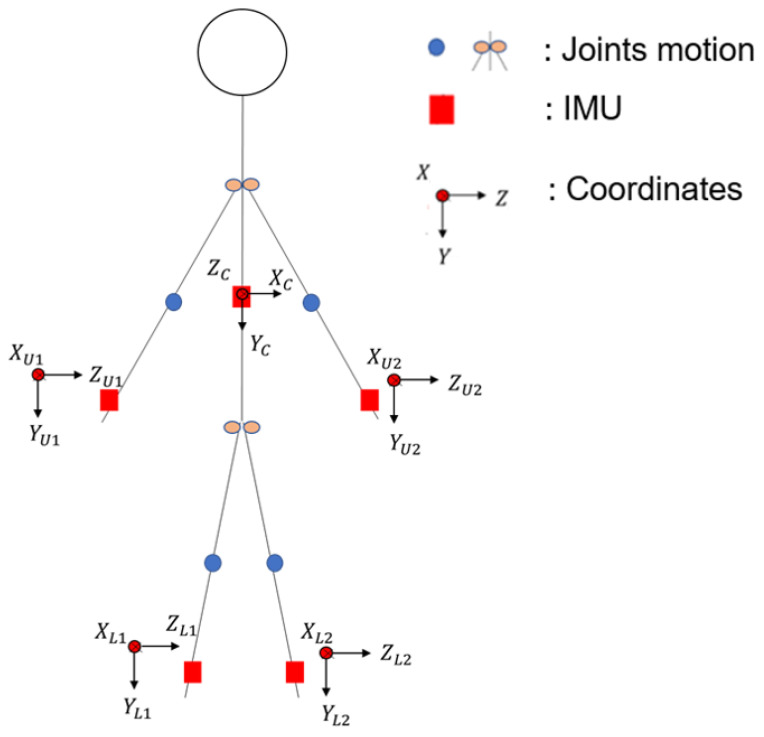
Minimal IMU sensor placement on body.

**Figure 7 sensors-24-04306-f007:**
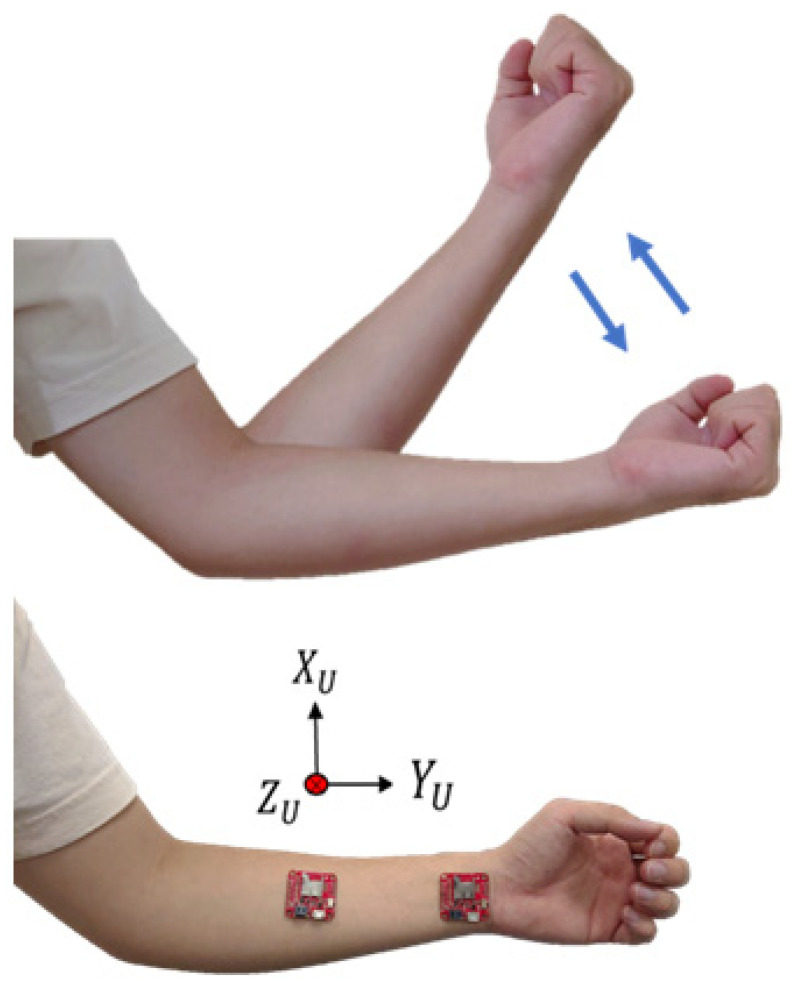
The position of the upper limb inertial sensors.

**Figure 8 sensors-24-04306-f008:**
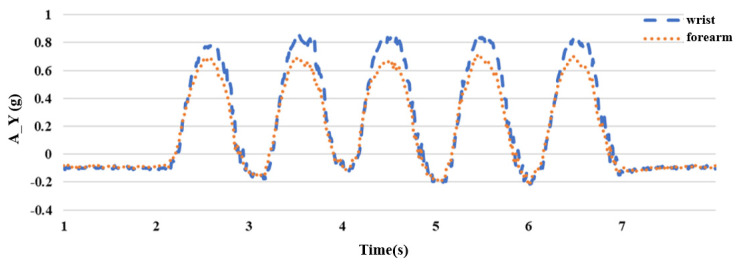
Comparison diagram of wrist and forearm *Y*-axis acceleration **A_Y**.

**Figure 9 sensors-24-04306-f009:**
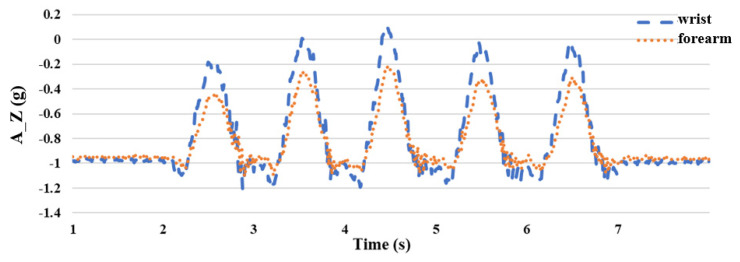
Comparison diagram of wrist and forearm *Z*-axis acceleration **A_Z**.

**Figure 10 sensors-24-04306-f010:**
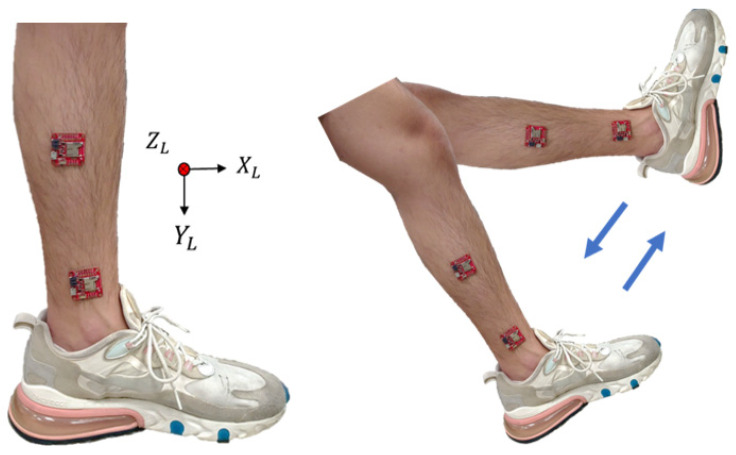
The position of the inertial sensor of the lower extremity.

**Figure 11 sensors-24-04306-f011:**
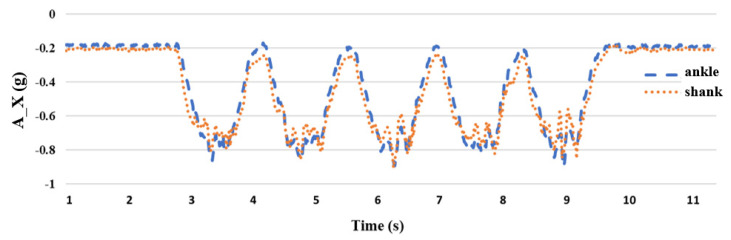
Comparison of *X*-axis acceleration **A_X** between the ankle and the shank.

**Figure 12 sensors-24-04306-f012:**
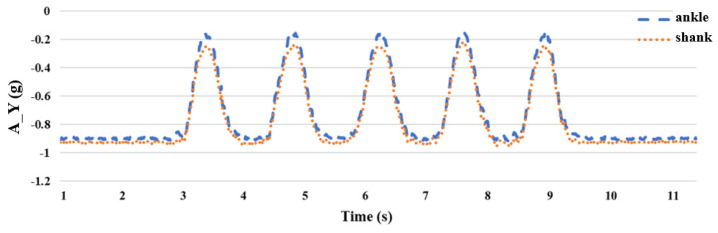
Comparison of *Y*-axis acceleration **A_Y** between the ankle and the shank.

**Figure 13 sensors-24-04306-f013:**
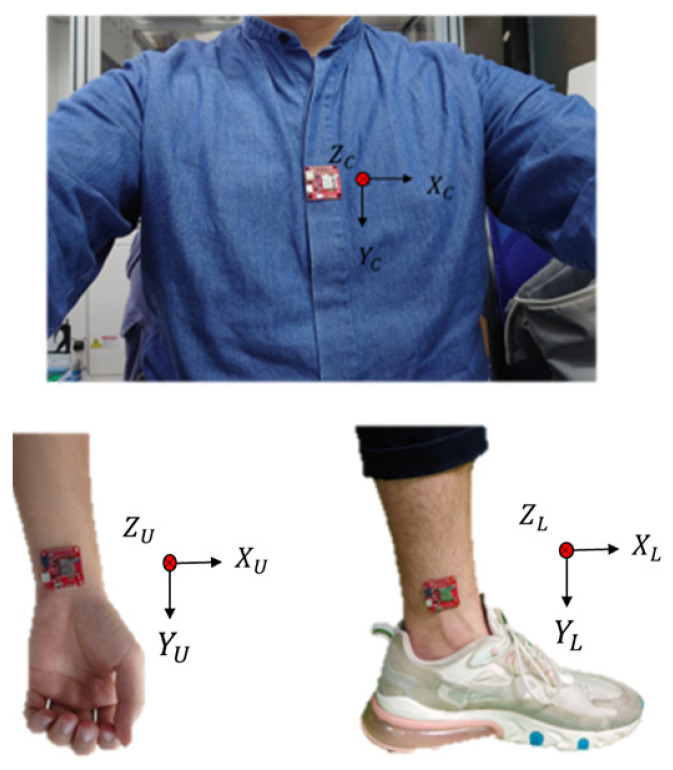
IMU placement and three-axis direction.

**Figure 14 sensors-24-04306-f014:**
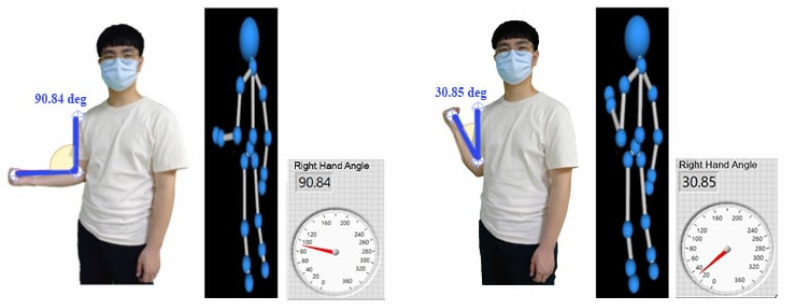
Kinect camera-based angle measurement for upper extremity positions.

**Figure 15 sensors-24-04306-f015:**
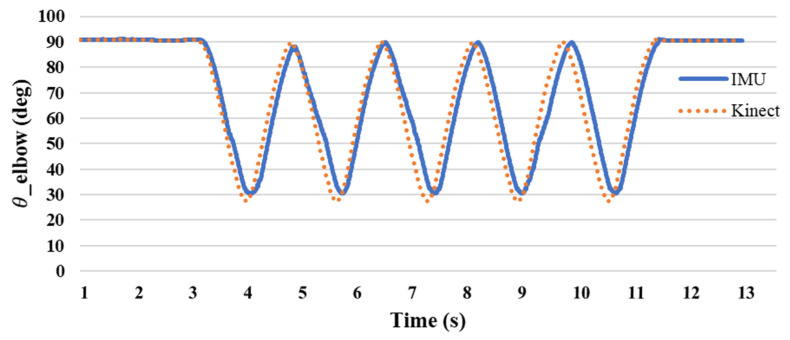
The difference between IMU and Kinect elbow joint angle.

**Figure 16 sensors-24-04306-f016:**
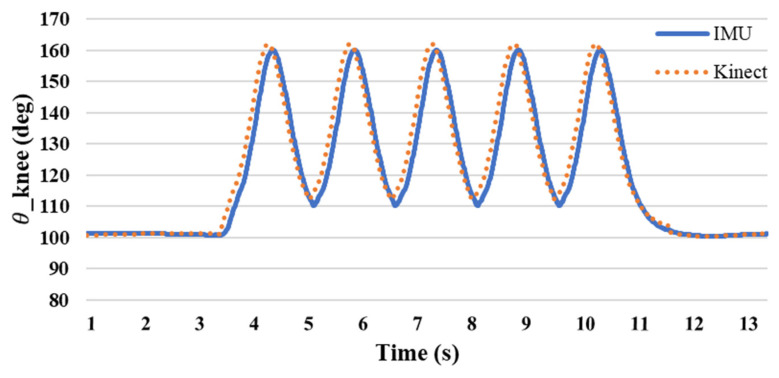
IMU and Kinect knee joint angle difference.

**Figure 17 sensors-24-04306-f017:**
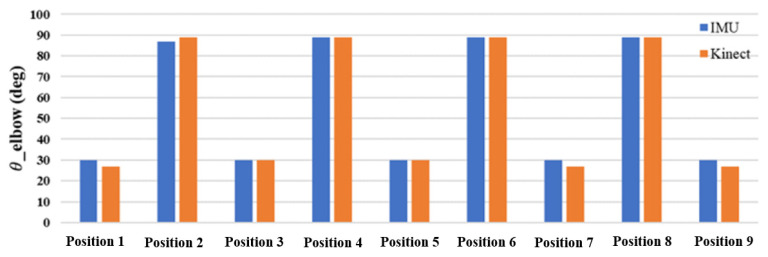
Comparison of elbow angles at various positions using IMU and Kinect.

**Figure 18 sensors-24-04306-f018:**
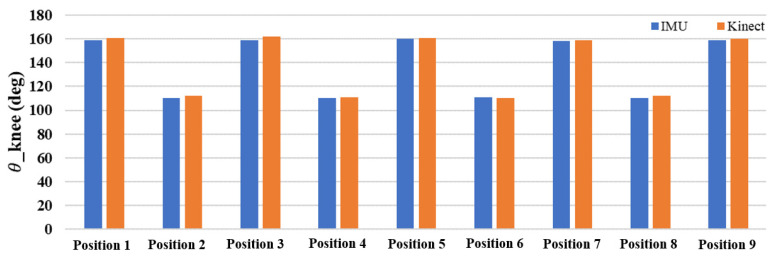
Comparison of knee angles at various positions using IMU and Kinect.

**Figure 19 sensors-24-04306-f019:**
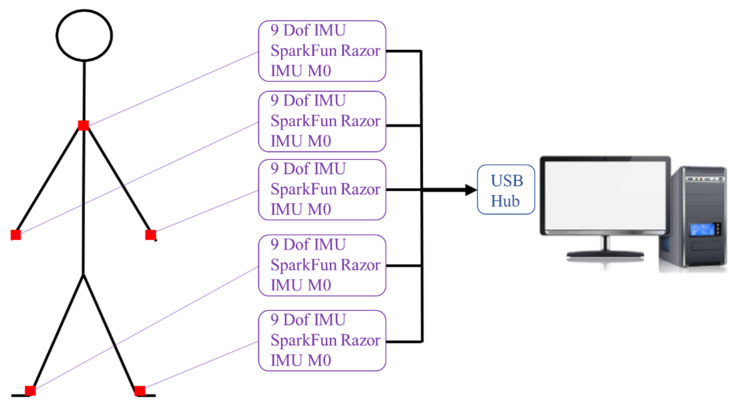
The hardware architecture of the new angle calculation system.

**Figure 20 sensors-24-04306-f020:**
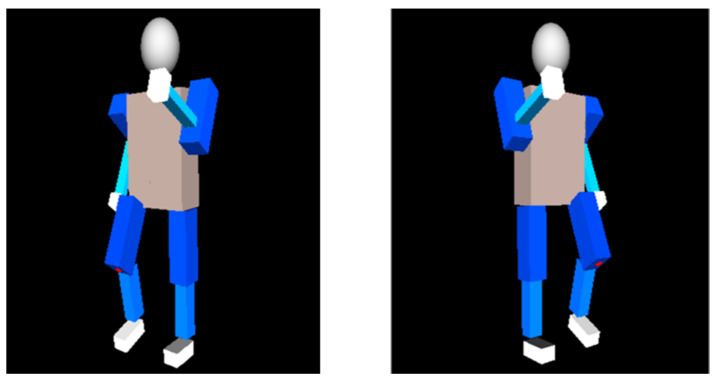
Virtual model walking motion posture simulation.

**Figure 21 sensors-24-04306-f021:**
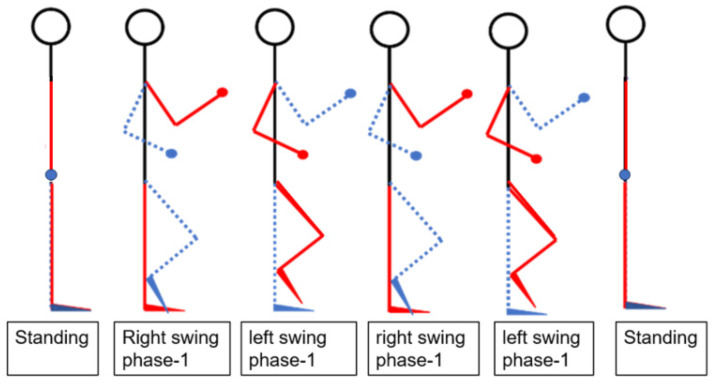
Schematic diagram of walking motion.

**Figure 22 sensors-24-04306-f022:**
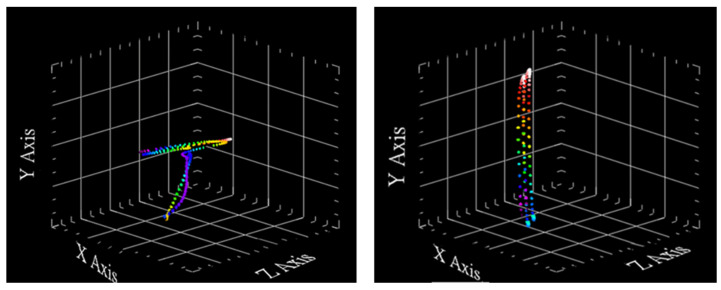
IMU trajectory diagram at the wrist and ankle during walking.

**Figure 23 sensors-24-04306-f023:**
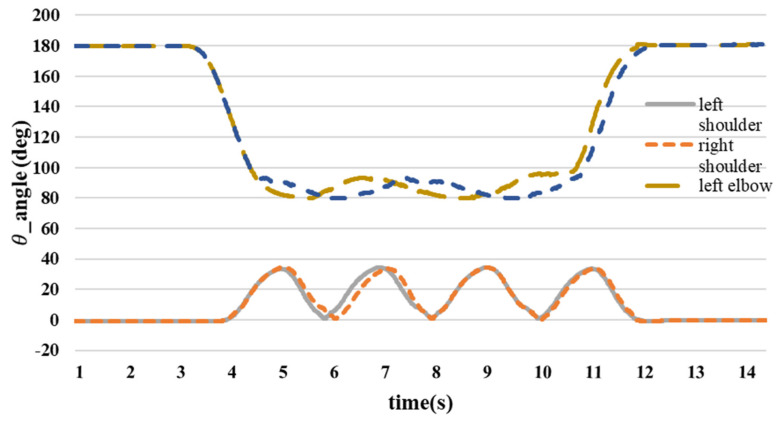
Upper limb joint angle.

**Figure 24 sensors-24-04306-f024:**
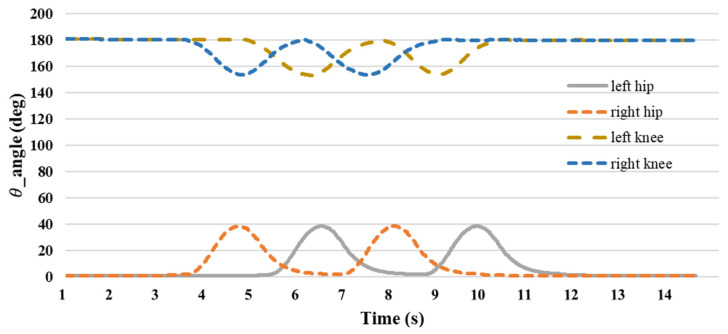
Lower extremity joint angle.

**Table 1 sensors-24-04306-t001:** Upper body code.

Symbol	Meaning
$l_{U1}$	Upper extremity length
$l_{U2}$	Forearm length
$\theta_{U1}$	Shoulder angle
*α* (pitch)	Calculated angles from the IMU
$\theta_{U3}$	Elbow angle
$A_{U},B_{U}$	The distance between the IMU coordinates and the origin
$x_{U},y_{U}$	The distance between the elbow joint coordinates and the origin

**Table 2 sensors-24-04306-t002:** Lower body code.

Symbol	Meaning
$l_{L1}$	Thigh length
$l_{L2}$	Shank length
$\theta_{L1}$	Hip angle
*beta* (pitch)	Calculated angles from the IMU
$\theta_{L3}$	Knee angle
$A_{L}$,$B_{L}$	The distance between the IMU coordinates and the origin
$X_{L}$,$Y_{L}$	The distance between the knee joint coordinates and the origin

**Table 3 sensors-24-04306-t003:** Joint angles involved in the motion of walking.

No		First Cycle		Second Cycle		Variance = ±2 Degree
Motion	Initial (Degree)	The Raising of the Left Hand and the Lifting of the Right Foot (Degree)	The Raising of the Right Hand and the Lifting of the Left Foot (Degree)	The Raising of the Left Hand and the Lifting of the Right Foot (Degree)	The Raising of the Right Hand and the Lifting of the Left Foot (Degree)	Finish (Degree)
Left shoulder	1.2	33.3	34.6	34.4	33.7	1.1
Right shoulder	1.4	34.1	33.7	34.7	33.5	1.3
Left elbow	180.2	82.5	92.4	83.8	94.6	180.4
Right elbow	180.5	90.8	85.2	81.3	81.8	180.6
left hip	1.1	2.4	38.7	2.4	38.5	1.2
Right hip	1.3	38.7	3.4	38.5	1.3	1.4
Left knee	180.3	180.7	153.3	179.6	153.4	180.5
Right knee	180.4	153.5	178.7	155.7	180.5	180.3

**Table 4 sensors-24-04306-t004:** Limitations and discussions of IMU-based posture recognition system.

Limitation	Description	Discussion
Number of IMU sensors	The study uses five sensors (two on the wrists, two on the ankles, and one on the chest) to track body movements.	Despite a reduction in the number of sensors compared to earlier research, the usage of five sensors might still provide challenges and inconveniences for users in practical applications. Addressing the difficulty of maintaining high precision while further reducing the number of sensors remains a priority.
Accuracy of joint angle estimation	The system demonstrates relatively high accuracy with discrepancies within 10° compared to dual IMU sensors and within 5° compared to image recognition technology, but there are still inherent errors.	This level of accuracy may not be enough for applications requiring very high precision, such as in the medical domain or professional sports. Improving the system’s precision is essential to expand its range of applications.
Cost and system complexity	The use of multiple sensors and complex algorithms, such as the extended Kalman filter, increases the system’s cost and complexity.	Optimizing cost and reducing system complexity are essential for broader commercial adoption.
Scalability and integration	The current system focuses on tracking joint angles of the upper and lower limbs during specific exercises like curling and leg stretching.	In order to have a wider range of uses, it is essential to enhance the system’s capacity to monitor intricate motions and incorporate with other systems, such as machine learning or artificial intelligence, for thorough data analysis.

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
