# Peer review of "Three-Dimensional Human Posture Recognition by Extremity Angle Estimation with Minimal IMU Sensor"

_sensors, 2024, doi:10.3390/s24134306_

Round 1
Reviewer 1 Report
Comments and Suggestions for Authors
This paper states its purpose is based on 3D posture recognition using a single inertial sensor and validates the work using a combination of a Kinect camera + dual IMU sensor setup. The system employs an Extended Kalman Filter to process data from the IMU's gyroscope, accelerometer, and magnetometer. A mathematical model based on spatial geometry is then used to estimate the shoulder, elbow, hip, and knee joint angles from the IMU data.
The research presents interesting findings from experiments determining optimal sensor placement for posture estimation. However, the paper would benefit from a clearer structure, beginning with explicitly stating the primary objective of using a single IMU sensor. It becomes a little confusing later when the work suggests multiple sensors, one per limb and one on the chest which seems to contradict the primary statement.
Some good results are shown but the paper lacks details on sample sizes and the range of movements recorded, which are essential for assessing the approach's robustness and generalizability.
Comments on the Quality of English Language
Some minor typos found when reading through:
Line 82: What is PD?
Line 85-96: Sentence is a little vague although intended meaning is still there. Could be written a bit better.
Line 138: “correction” starts with lower case whereas previously and later, it is upper case.
Line 399: table 3 should be “Table 3”
Author Response
Thank you for your comments!
I have answered your comments in the Word file and highlighted them in the manuscript
Chinese-English Dictionary Enable Select Search My Words

Reviewer 2 Report
Comments and Suggestions for Authors
Brief summary: The aim of the study is to present a novel system for the identification of the three-dimensional motion angles and postures of both the human upper and lower limbs by using a single inertial measurement unit and a spatial geometric equation. The main advantage of this wearable system is the continuous monitoring of body movements without classical issues associated to camera-based methods, i.e. spatial limitations or occlusion.
Major comments:
The topic of this work is original, causing a probable interest in the reader. However, several changes are needed to improve the clarity of presentation and the comprehensibility of the study.
1. The novelty of the work is well stressed, but the first part of the introduction should be rearranged, because the flow is not so clear. I suggest introducing the concept of human posture recognition in general, moving to traditional motion capture technologies such as Kinect, and then introducing the advantages of IMUs.
2. Even if the functioning of Kalman Filter is described in reference 26, to highlight the readability of Figure 2 it is better to briefly explain which are the terms in every block.
3. Why in Figures 8 and 9 the label of y axis contains “9.8 m/s2” as the measure unit? Same consideration for Figures 12 and 13.
4. Which is the content of Figures 17 and 18? Do they contain medium values of the entire acquisition? If yes, can you explain it in the text and add the standard deviation values on the bars?
5. What colors in Figure 22 mean?
6. Measure units are missing in Table 3.
7. Limitations of this study should be listed and discussed.
Minor comments:
- Lines from 85 to 88: please add a subject to this sentences
- Lines 137 and 140: remove the capital letter from the term “correction”
Comments on the Quality of English LanguageA deep check of English should be made, especially considering the alternance of past and presence tenses in the text.
Author Response

(The authors gave the same response as above.)

Reviewer 3 Report
Comments and Suggestions for Authors
The article is interested, but the authors do not have enough knowledge of the human body nomenclature and knowledge of human body measurements. After improvements I included the article will be OK.

Author Response

(The authors gave the same response as above.)

Round 2
Reviewer 2 Report
Comments and Suggestions for Authors
I would like to thank the authors for their dedication in satisfying my requests.
I still have doubts about the bar diagrams in Figures 17-18: what first, second, etc stand for? And also about the colours in Figure 22, which are not clear despite the explanation. Once these small requirements have been met, the paper can be accepted.
Comments on the Quality of English LanguageEnglish has been improved.
Author Response
Dear Reviewer,
Thank you for your valuable comments. I have addressed your comments in the Word file and highlighted them in the manuscript.
Chinese-English Dictionary Enable Select Search My Words
